# Decoding neural responses to temporal cues for sound localization

**Dan FM Goodman[1,2]\*[†a†b], Victor Benichoux[3,4], Romain Brette[3,4]\***

[1]Laboratoire de Psychologie de la Perception, CNRS and Université Paris Descartes, Paris, France; [2]Département d'Etudes Cognitives, Ecole Normale Supérieure, Paris, France; [3]Laboratoire de Psychologie de la Perception, CNRS and Université Paris Descartes, Paris, France; [4]Département d'Etudes Cognitives, Ecole Normale Supérieure, Paris, France

**Abstract** The activity of sensory neural populations carries information about the environment. This may be extracted from neural activity using different strategies. In the auditory brainstem, a recent theory proposes that sound location in the horizontal plane is decoded from the relative summed activity of two populations in each hemisphere, whereas earlier theories hypothesized that the location was decoded from the identity of the most active cells. We tested the performance of various decoders of neural responses in increasingly complex acoustical situations, including spectrum variations, noise, and sound diffraction. We demonstrate that there is insufficient information in the pooled activity of each hemisphere to estimate sound direction in a reliable way consistent with behavior, whereas robust estimates can be obtained from neural activity by taking into account the heterogeneous tuning of cells. These estimates can still be obtained when only contralateral neural responses are used, consistently with unilateral lesion studies.

**\*For correspondence:**
dan_goodman@meei.harvard.edu
(DFMG); romain.brette@ens.fr (RB)

[†]**Present address:** [a]Eaton Peabody Laboratory, Massachusetts Eye and Ear Infirmary, Boston, United States; [b]Department of Otology and Otolaryngology, Harvard Medical School, Boston, United States

**Competing interests:** The authors declare that no competing interests exist.

**Reviewing editor**: Dora Angelaki, Baylor College of Medicine, United States

## Introduction

To localize sound sources in the horizontal plane, humans and many other species use submillisecond timing differences in the signals arriving at the two ears (*Ashida and Carr, 2011*). The ear closer to the source receives the sound earlier than the other. These interaural time differences (ITDs) are encoded in the auditory brainstem by binaural neurons, which are tuned to both frequency and ITD. An influential theory proposes that ITD is represented by the activity pattern of cells with heterogeneous tunings, a pattern code for sound location (*Jeffress, 1948*). In a stronger version, ITD is represented by the identity of the most active cell in each frequency band, a labeled line code for sound location. Although this theory has proved successful in barn owls (*Konishi, 2003*), discrepancies have been observed in mammals. In particular, at low frequencies, many cells have best delays (BDs) larger than the physiological range of ITDs experienced by the animal (*McAlpine et al., 2001*). In a labeled line code, these cells would not have any function. An alternative theory was proposed, in which ITD is coded not by the firing of individual cells, but by the relative summed activity of each hemisphere, a summation code for sound location (*Stecker et al., 2005*; *Grothe et al., 2010*).

The nature of the neural code for ITD in mammals is still contentious because it is not known whether the auditory system sums activity or uses cell identity in decoding responses. In favor of the summation code hypothesis, cells with large BDs maximize ITD sensitivity of firing rate in the physiological range, whereas they are useless in a labeled line code. However, most of the cells with BDs inside the physiological range (most cells in cats; *Kuwada and Yin, 1983*; *Yin and Chan, 1990a*) actually degrade a summation code because their rates do not vary monotonically with ITD.

In simple situations where there is a single acoustical stimulus (e.g., tone) with unknown ITD, theoretical arguments show that a summation code is optimal at low frequencies (*Harper and McAlpine, 2004*).

**eLife digest** Having two ears allows animals to localize the source of a sound. For example, barn owls can snatch their prey in complete darkness by relying on sound alone. It has been known for a long time that this ability depends on tiny differences in the sounds that arrive at each ear, including differences in the time of arrival: in humans, for example, sound will arrive at the ear closer to the source up to half a millisecond earlier than it arrives at the other ear. These differences are called interaural time differences. However, the way that the brain processes this information to figure out where the sound came from has been the source of much debate.

Several theories have been proposed for how the brain calculates position from interaural time differences. According to the hemispheric theory, the activities of particular binaurally sensitive neurons in each of side of the brain are added together: adding signals in this way has been shown to maximize sensitivity to time differences under simple, controlled circumstances. The peak decoding theory proposes that the brain can work out the location of a sound on the basis of which neurons responded most strongly to the sound.

Both theories have their potential advantages, and there is evidence in support of each. Now, Goodman et al. have used computational simulations to compare the models under ecologically relevant circumstances. The simulations show that the results predicted by both models are inconsistent with those observed in real animals, and they propose that the brain must use the full pattern of neural responses to calculate the location of a sound.

One of the parts of the brain that is responsible for locating sounds is the inferior colliculus. Studies in cats and humans have shown that damage to the inferior colliculus on one side of the brain prevents accurate localization of sounds on the opposite side of the body, but the animals are still able to locate sounds on the same side. This finding is difficult to explain using the hemispheric model, but Goodman et al. show that it can be explained with pattern-based models.

Previous studies have also shown that with simple stimuli, taking into account cell identity rather than simply summing all responses does not improve performance (*Lesica et al., 2010*; *Lüling et al., 2011*). However, what is optimal in a simple world may not be optimal in an ecological environment. In a simple situation where only the ITD varies, the optimal code is the most sensitive one. In complex situations where other dimensions also vary, there is a trade-off between sensitivity and robustness, so the optimal code is not the most sensitive one (*Brette, 2010*). In fact, theory predicts that in complex situations, the heterogeneity of ITD tunings is critical to produce robust estimates.

To address this, we studied the performance of different decoders in increasingly complex situations, including variations in spectrum, background noise, and head-related acoustic filtering. We found that summing cell responses is strongly suboptimal and that heterogeneity in tunings is information rather than noise.

## Results

### Decoding the sound's ITD from cell responses

Previous studies have tested the performance of simple decoders based on single-unit cell responses to acoustical stimuli (*Fitzpatrick et al., 1997*; *Hancock and Delgutte, 2004*; *Stecker et al., 2005*; *Devore et al., 2009*; *Miller and Recanzone, 2009*; *Lesica et al., 2010*; *Lüling et al., 2011*). However, this approach is limited to a small number of acoustical stimuli and cells. Here, we wanted to test the performance of different decoders based on the response of a large population (up to 480 cells) to a large variety of sounds totaling 11 hr of sound per cell. Obtaining this amount of data from electrophysiological recordings is not feasible because it would correspond to more than 7 months of single-unit recordings. We therefore decided to base our comparison on responses generated by a standard computational model, fitted with empirical data, which has been shown to produce realistic responses ('Materials and methods').

First, we sampled many cells from a distribution of BD vs best frequency (BF) (*Figure 1A*, left). For guinea pigs, the distribution was defined by the measured mean and variance of BD as a function of BF (*McAlpine et al., 2001*). For cats, we fitted a distribution to a set of measurements of BDs and BFs

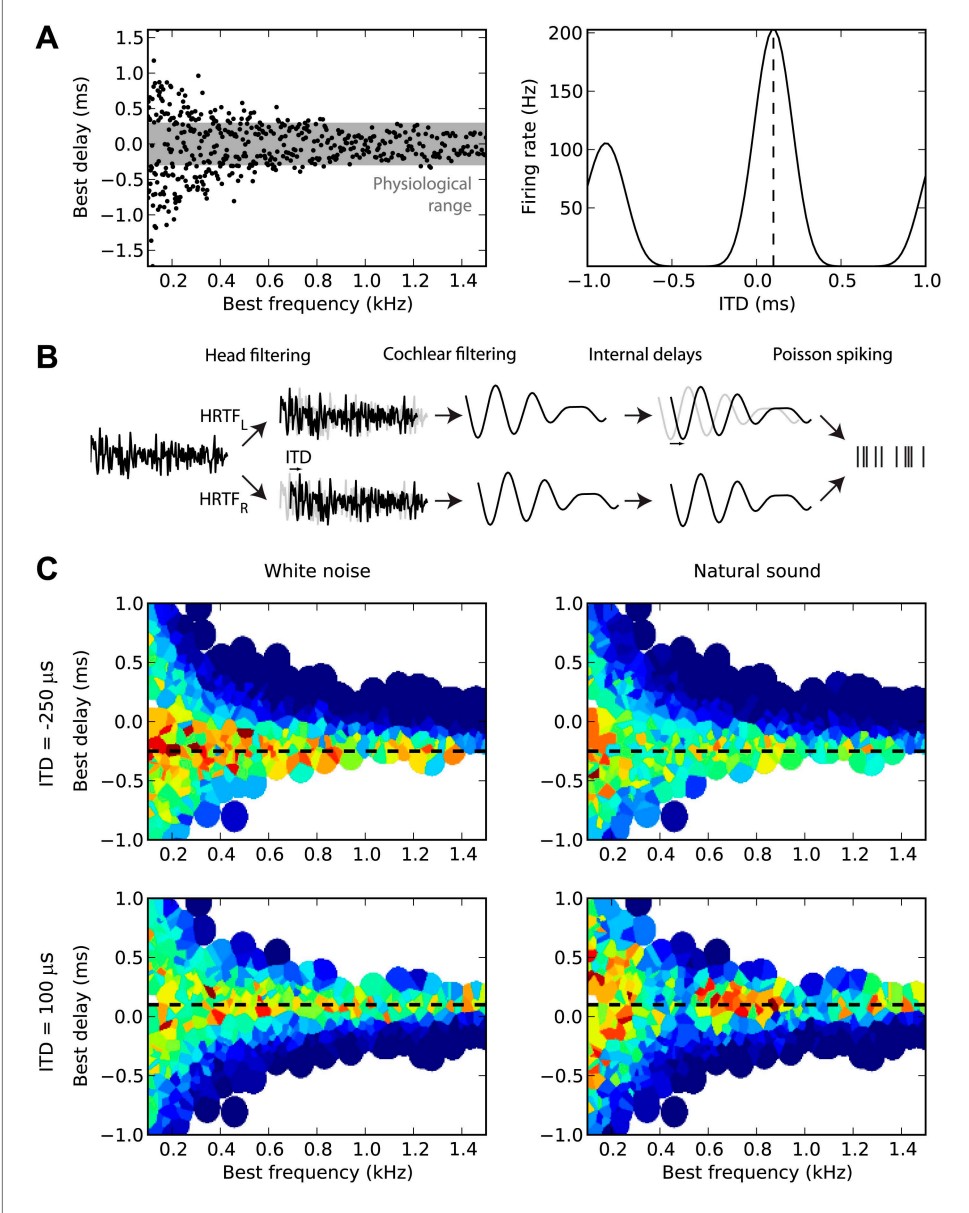

**Figure 1**. Overview of model. (**A**) The distribution of best delay vs best frequency for cells in the guinea pig model (left), with the physiological range of ITDs shown in gray, and a sample tuning curve (right). (**B**) Illustration of the model: a sound source is filtered via position-dependent HRTFs to give left and right channels. For each best frequency on each channel, the signal undergoes cochlear filtering using gammatone filters. An internal delay is added, and the two channels are combined and sent to a binaural neuron model that produces Poisson distributed spikes. (**C**) The response of the cells to sounds at two different ITDs (rows) for white noise (left column) and a natural sound (right column). The ITD is indicated by the black dashed line. Each cell is surrounded by a disc with a color indicating the response of that neuron (hot colors corresponding to strong responses). When two or more discs overlap, each point is colored according to the closest cell. The strongest responses lie along the line of the ITD.

in 192 cells (the source data was in terms of characteristic frequency rather than BF, but these are equivalent for the linear model used here) (*Joris et al., 2006*). The cells are then modeled as generalized cross-correlators with an internal delay BD (*Yin et al., 1987*) (*Figure 1A*, right). *Figure 1B* illustrates the details of the model. We first model the acoustical propagation of the sound to the two ears. In the first part of this study, we consider only fixed ITDs, ignoring diffraction effects. In the second part, we move toward more realistic cues, using measured head-related transfer functions (HRTFs).

The signals are then band-pass filtered around the cell's BF using gammatone filters and normalized (representing the saturation seen in bushy cells; *Kuenzel et al., 2011*) and then crosscorrelated with an internal delay equal to the cell's BD ('Materials and methods'). The result is the firing rate of the cell, and we generate spikes with Poisson statistics. *Figure 1C* displays the responses of 480 cells of the guinea pig model to white noise (left column) and to a natural sound (right column) at two different ITDs (top and bottom). We will then estimate the sound's ITD from these population responses, using various decoders.

*Figure 2A* illustrates the peak and hemispheric decoders. A 100-ms sound is presented at 200 µs ITD. The peak decoder picks the most active cell and reports its BD as the estimated ITD. We observe already that although we chose cells with BFs in a narrow frequency band (640–760 Hz), the peak decoder performs poorly because of the noise in spiking. Therefore, we introduce a smoothed peak

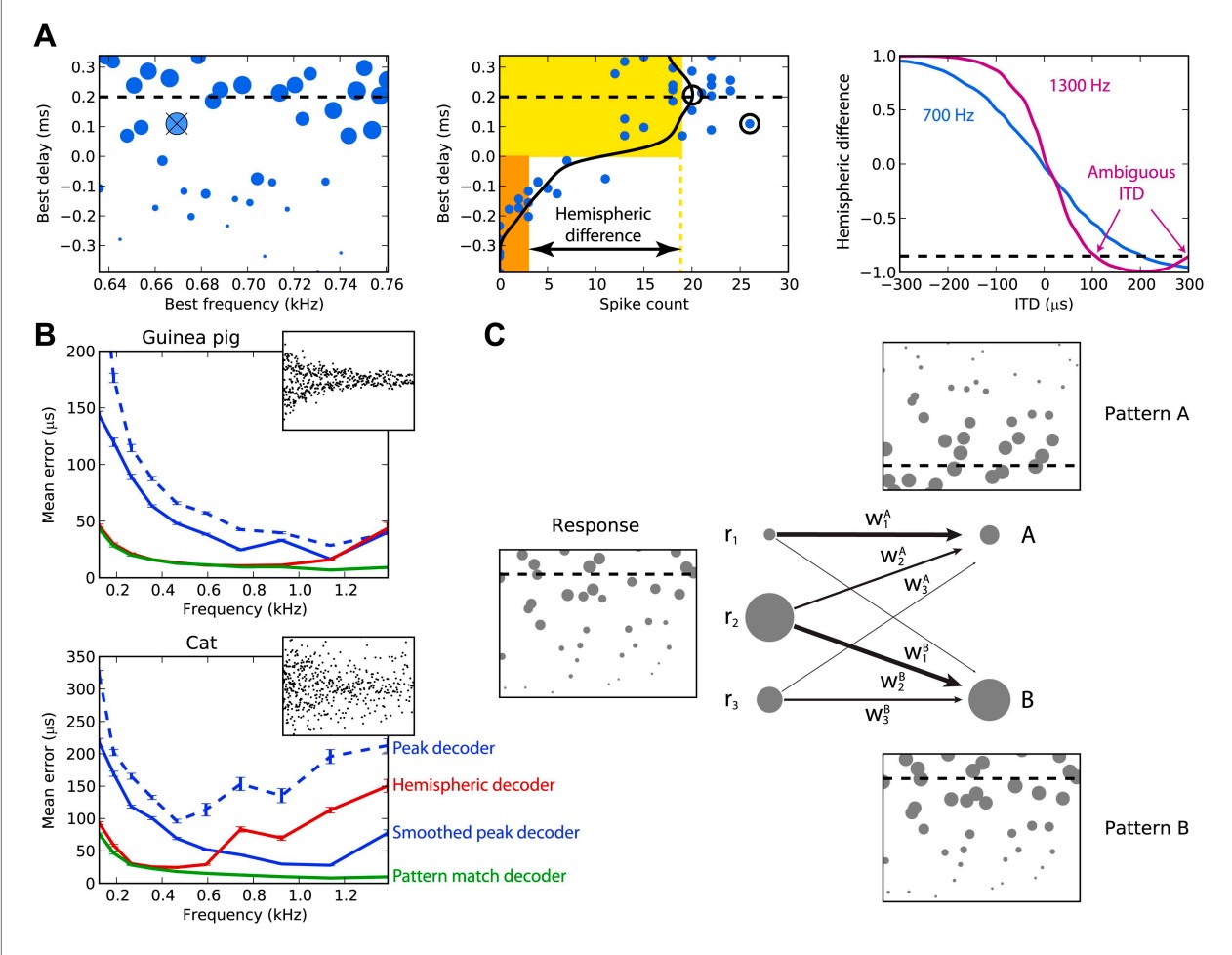

**Figure 2**. Decoders in single frequency bands. (**A**) Peak and hemispheric decoders. Left: response of binaural neurons to sound at ITD = 0.2 ms (dashed line), in a narrow frequency band. The size of points is proportionate to spike count, and the crossed point corresponds to the highest spike count. Middle: the same cell responses are displayed as best delay vs spike count (note the different horizontal axis). The solid black line is the Gaussian smoothed spike count, whose peak (circle) is the ITD estimate. The maximally responsive neuron is also indicated with a circle for comparison. The yellow and orange bars give the mean response of neurons with positive and negative best delays, respectively, from which the normalized hemispheric difference is computed. Right: the hemispheric difference as a function of ITD at 700 Hz (blue) and 1.3 kHz (purple). At 1.3 kHz, the difference shown by the dashed line gives an ambiguous estimate of the ITD. (**B**) Mean error for the guinea pig and cat, for the peak (blue, dashed), smoothed peak (blue, solid), hemispheric (red), and pattern match (green) decoders. The distribution of BD vs BF is shown in the inset. (**C**) Illustration of the pattern match decoder and a neural circuit that implements it. The response (left) is compared to two patterns A and B, corresponding to two different ITDs (right). Each response neuron is connected to a pattern-sensitive neuron with weights proportional to the stored response of each pattern. When the weights match the responses, the output of the pattern-sensitive neuron is strongest.

decoder. We first discard the information about BF, and simply consider the spike count of cells as a function of their BD. This relationship is smoothed to reduce noise, and we take the BD at the peak of the smoothed curve (one of the possible variations of crosscorrelation models; *Stern and Trahiotis, 1995*). Smoothing could be neurally implemented by pooling the activity of cells with similar tuning. This decoder is less noisy. Finally, we consider the hemispheric decoder, in which we also discard information about BD in each hemisphere. To simplify, we define each hemisphere as the set of cells with BDs of the same sign. We calculate the total spike count for each hemisphere (yellow and orange rectangles) and compute the difference, normalized by the total activity. This gives a value between −1 and 1, the hemispheric difference, which varies systematically with ITD (*Figure 2A*, right). Therefore, from the observation of the difference, one can invert the relationship and infer the ITD. Note, however, that this relationship depends on the frequency band in which the hemispheric difference is computed. In blue, cells are picked with BFs around 700 Hz and the hemispheric difference varies almost linearly with ITD. In purple, cells are picked with BFs around 1300 Hz and the curve is a sigmoid. More importantly, ambiguities start to occur at these high BFs: for example, a hemispheric difference of −0.8 is consistent with ITDs of both 100 and 300 µs. This occurs when the physiological range of ITD represents more than one period of the auditory filter's center frequency.

We now systematically test the estimation error of these three decoders for cells whose BFs are in narrow frequency bands within the range 100–1500 Hz (*Figure 2B*). Stimuli are white noise bursts lasting 100 ms. For the hemispheric decoder, we use the hemispheric difference curve calculated in the same frequency band in which it is tested. Thus, there is a specific decoder for each frequency band tested, which is the most favorable scenario. As expected, the peak decoder performs very poorly, both for the guinea pig and the cat models. The two animal models differed by the BD distributions and by the physiological range of ITDs (300 µs for the guinea pig model, *Sterbing et al., 2003*; 450 µs for the cat model, *Tollin and Koka, 2009a*). Using the smoothed peak decoder improves substantially on these results. The hemispheric decoder performs better than both decoders for the guinea pig model at all frequencies, but for the cat, it is only better than the smoothed peak decoder for frequencies below 600 Hz. Thus, it appears that even for this simple scenario, the hemispheric decoder is a very poor decoder of ITD for the cat model, except at very low frequency. The fact that the estimation error of the hemispheric decoder starts increasing at a lower frequency for the cat than for the guinea pig model was expected based on the larger head size of the cat (*Harper and McAlpine, 2004*).

The reasons for the limitations of the different decoders are simple. Because the peak decoder selects a single cell, its estimation error reflects the level of noise in individual responses, which is high. The smoothed decoder improves on this matter, but still mostly uses the responses of cells with similar tuning. In addition, at low frequencies, both estimators rely on the responses of a small pool of cells with BDs inside the physiological range. The hemispheric decoder sums all responses, which reduces noise but also discards all information about BF and BD.

We introduce the pattern match decoder, a simple decoder that addresses both problems (*Figure 2C*). We calculate the average response of cells to sounds presented at each ITD to be identified. This population response, which we call a pattern, is stored in a vector ($w_1$, …, $w_n$), normalized to have length 1. When a sound is presented, the cell responses are compared with the patterns by computing a normalized dot product between the responses and the patterns, varying between 0 (perfectly dissimilar) and 1 (perfectly similar) ('Materials and methods' for formulae). This can be implemented by a single-layer neural network in which the output neurons encode the preferred ITD and the synaptic weights represent the patterns. The reported ITD is the ITD associated to the most similar pattern, that is, with the highest dot product.

*Figure 2B* also shows the performance of the pattern matching decoder. As for the hemispheric decoder, patterns were computed in the same frequency band in which the decoder is tested. The pattern match decoder performs better than both the other decoders, both for the guinea pig and the cat models. The difference with the hemispheric decoder is very large for the cat model, but for the guinea pig model, it only starts being substantial above 1 kHz. The pattern match decoder combines the advantages of the hemispheric and peak decoders: it averages spiking noise over all cells, but it still uses individual information about BF and BD. The purpose of introducing this decoder is not to suggest that the auditory system extracts information about sound location in this exact way, but rather to estimate how much information can be obtained from the heterogeneous responses of these neurons. We also tested several other standard decoders from machine learning, including optimal linear decoding, maximum likelihood estimation, and nearest neighbor regression, but the pattern

match decoder outperformed them in all cases and so we do not present the results of these decoders here (although see *Figure 3—figure supplement 1* for a sample of these results).

## Integrating information across frequency

The estimation task in *Figure 2* was very simple because we trained and tested the decoders in the same narrow frequency bands. In *Figure 3*, we investigate the issue of frequency integration. All decoders are now trained with white noise at various ITDs, considering all cells with BFs between 100 Hz and 1.5 kHz. For the hemispheric decoder, this means that we pool the responses of all cells in the same hemisphere, for all BFs, and we use a single broadband hemispheric difference curve to estimate ITDs. Decoder performance is then tested with white noise. For this more realistic task, it appears that the error made by the pattern match decoder is about half the error of the hemispheric decoder for the guinea pig model. For the cat models, this difference is even larger. In fact, it turns out that for the cat, the smoothed peak decoder performs better than the hemispheric decoder. To understand why, we now test the decoders on band-pass noises, as a function of the center frequency, while the decoders are still trained with broadband noise (*Figure 3B*). This test addresses the robustness of these decoders to changes in sound spectrum. We make two observations. First, all decoders perform much worse than when decoders are trained and tested in the same frequency bands (compare with *Figure 2B*; the unsmoothed peak decoder performs very poorly and is not shown). This means that frequency integration is indeed an issue. Second, the hemispheric decoder performs worse than the two other decoders above 700 Hz for the guinea pig models and above 500 Hz for the cat. This was expected for two reasons: (1) the hemispheric difference is ambiguous at high frequency (above about 1200 Hz for both animals), and (2) the hemispheric difference depends not only on ITD but also on frequency (*Figure 2A*, right). We attempt to solve the first problem by discarding all cells with BF higher than a specified cutoff frequency (*Figure 3C*). Performance is tested again with white noise. Both for the guinea pig and the cat models, the error of the hemispheric decoder starts increasing when cells with BF above 1.2 kHz are included. For this reason and because the hemispheric difference becomes ambiguous above 1.2 kHz in both models, we restrict to cells with BF <1.2 kHz in the rest of

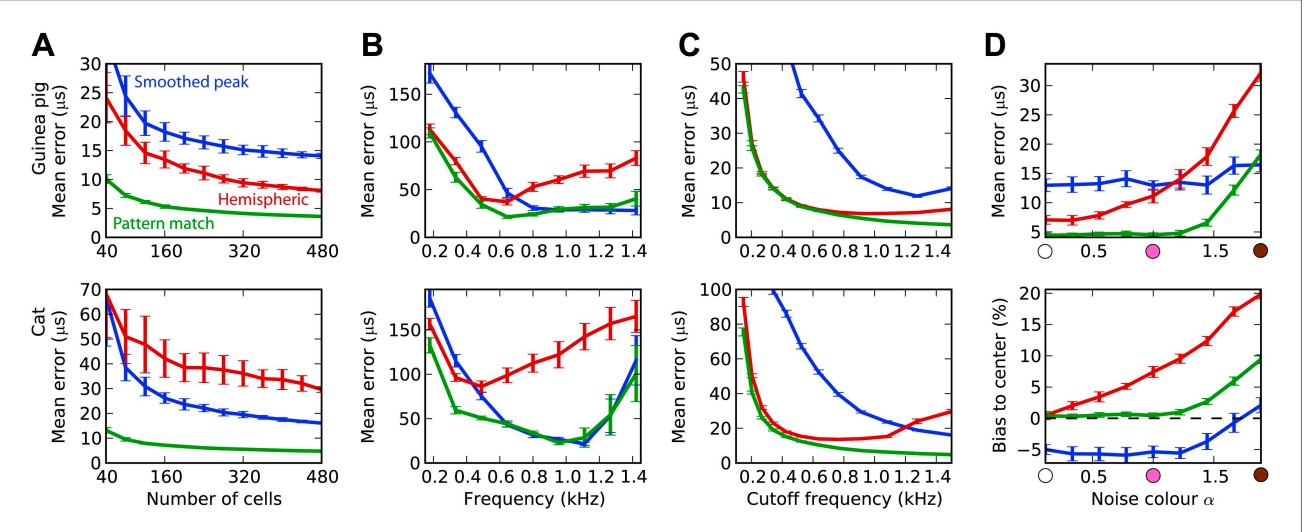

**Figure 3**. Integration across frequencies. (**A**) Mean error in estimating ITD for white noise using the smoothed peak (blue), hemispheric (red), and pattern match (green) decoders, as a function of the number of binaural cells. Training and testing of the decoders are both performed using white noise. (**B**) Mean error as a function of frequency band when decoders are trained on white noise but tested on band-pass noise centered at the given frequency. Notice the different vertical scale between (**A** and **B**). (**C**) Performance when cells with a frequency above the cutoff are discarded. (**D**) Mean error and bias to center in the decoders for guinea pig (with a maximum frequency of 1.2 kHz) when trained on white noise and tested on colored noise.

The following figure supplements are available for figure 3:

**Figure supplement 1**. Comparison with standard machine learning decoders.

this study. We note, however, that the error of the pattern match decoder continues decreasing as cells with high frequency are added.

We have seen that estimation error depends on the frequency band of the presented sound (*Figure 3B*). This observation extends to broadband sounds that vary in spectrum. We tested the estimation performance for the guinea pig models when the decoders are trained on white noise and tested with $1/f^\alpha$ noise: from white noise ($\alpha = 0$) to pink ($\alpha = 1$) and brown ($\alpha = 2$) (*Figure 3D*). The hemispheric decoder is not robust to changes in spectrum, as the error increases with noise color $\alpha$. The pattern match decoder shows the same trend, but the error remains constant on a larger range. The error of the smoothed peak decoder does not depend on sound spectrum. The main reason for the lack of robustness of the hemispheric decoder is shown in *Figure 3D* (bottom). As $\alpha$ increases, the estimate of the ITD becomes more and more biased to the center. This is because with high $\alpha$, every cell receives more low frequencies than high frequencies compared to the white noise case, and therefore the hemispheric difference curve changes and becomes flatter (*Figure 2A*, right). Note that this happens not by the recruitment of more low-frequency cells but also by the change in the hemispheric difference for all cells.

We now attempt to improve frequency integration in the hemispheric decoder by taking into account the change in hemispheric difference with frequency (*Figure 4A*). The ITD tuning curves have different shapes, depending on the cell's BF (left). As a result, the hemispheric difference in each frequency band varies with the center frequency (middle). The curves are shallower in low frequency and sharper in high frequency (right). In fact, the slope is expected to be proportional to frequency:

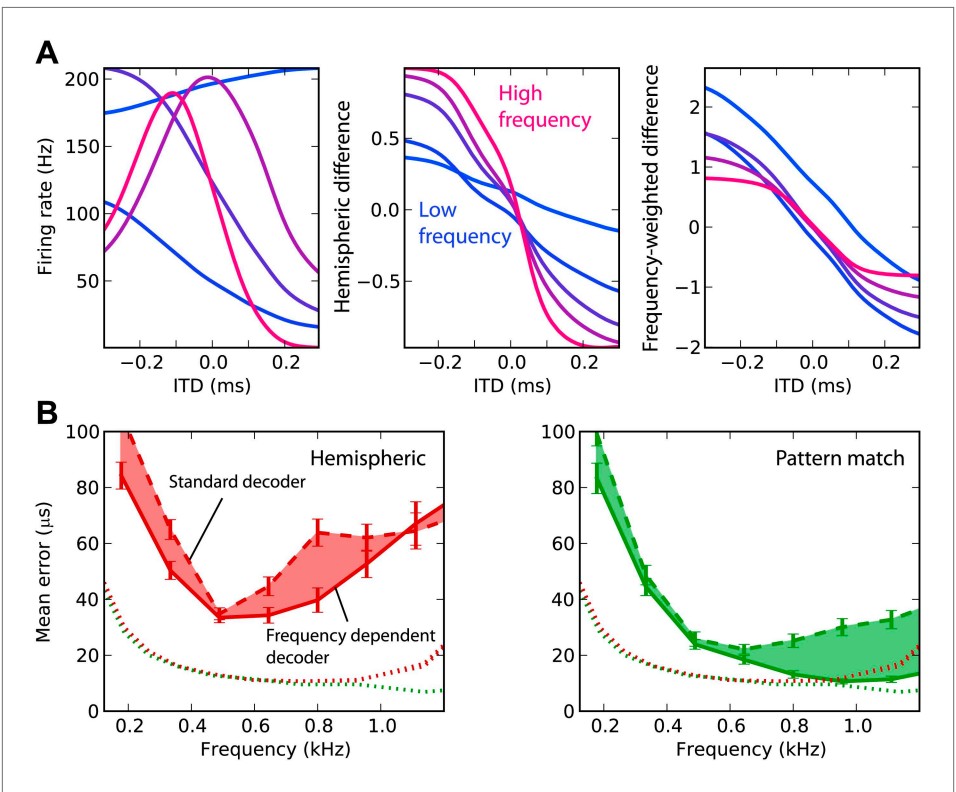

**Figure 4**. Frequency-dependent improvements. (**A**) Comparing hemispheric differences across frequency channels. In each plot, color indicates frequency with red being high frequency and blue being low frequency. Left: tuning curves for a few binaural neurons. Middle: hemispheric difference $(L − R)/(L + R)$. Right: frequency-dependent hemispheric difference $(1/f)$ $(L − R)/(L + R)$. (**B**) Mean error as a function of frequency band in the guinea pig model, for hemispheric (red) and pattern match (green) decoders (dashed lines), and frequency-dependent hemispheric and pattern match (solid) decoders. The shaded regions show the difference between the simple and frequency-dependent versions. The dotted lines show the mean error for band-pass noise if the decoder is trained and tested on the same frequency band, as shown in *Figure 2B* (guinea pig). This represents the lower bound for the error.

the hemispheric difference is determined by the interaural phase difference, which is the product of frequency and ITD. Therefore, we fix this issue by normalizing the cell responses by their BF in the calculation of the hemispheric difference ('Materials and methods'). This produces hemispheric differences with similar slopes in all frequency bands. Note that there are constant biases due to the fact that the cells' BDs are not exactly symmetrical between the two hemispheres (only their distribution is).

This frequency-dependent correction indeed improves ITD estimation when the decoder is trained on broadband noise and on band-passed noise (*Figure 4B*). However, the error still remains higher than in the simple case when the decoder is trained and tested in the same frequency band. In the same way, we improved frequency integration for the pattern match decoder by calculating intermediate estimates in each frequency band and combining the results ('Materials and methods'). This correction improves the performance above 600 Hz, where it is close to the performance obtained in the simple case. In the remainder of this study, we only consider these two frequency-corrected decoders.

## Background noise and sound diffraction

We then test the decoders in increasingly realistic situations. First, we consider the effect of background noise on performance (*Figure 5*). Interaural correlation is decreased by adding dichotic background noise to the binaural signals, and the estimation error is computed as a function of signal-to-noise ratio (SNR). All decoders were trained in quiet. In all cases, the pattern match decoder performs best, but

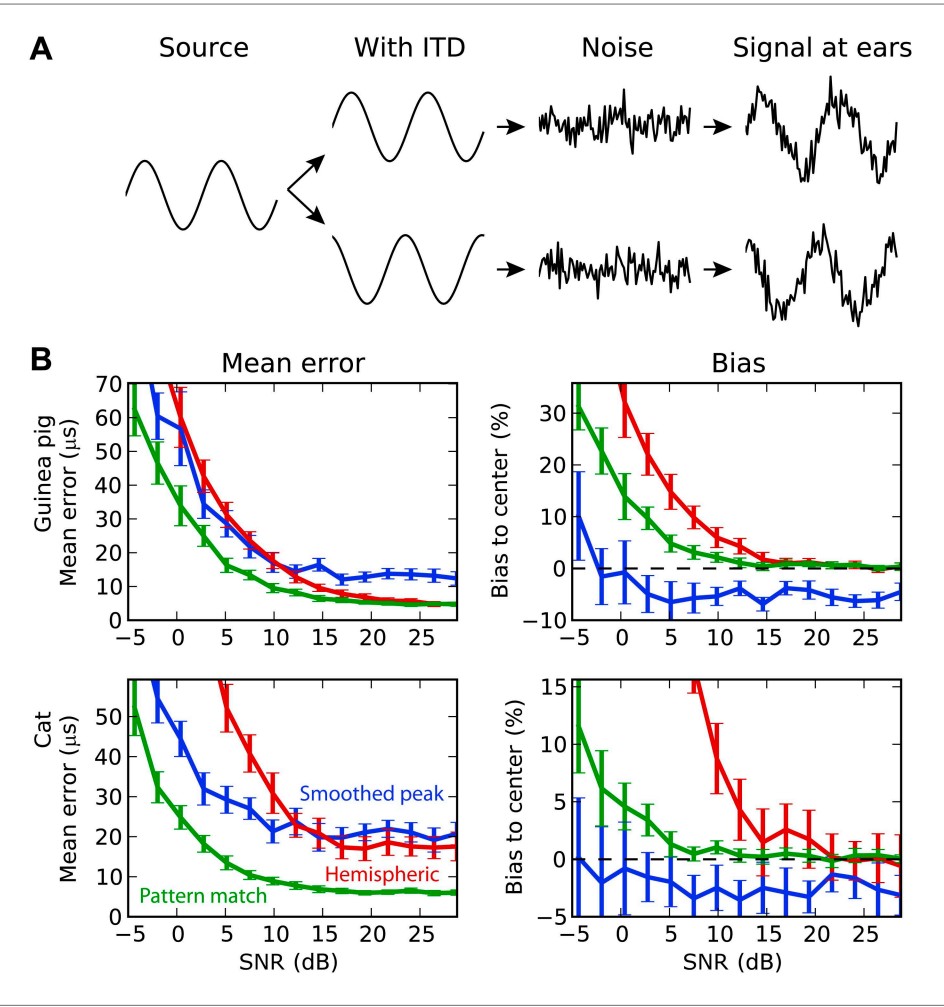

**Figure 5**. Background acoustic noise. (**A**) Illustration of protocol: a binaural sound is presented with a given ITD, with independent white noise added to the two channels. (**B**) Mean error (left column) and central bias (right column) at different signal to noise levels. Decoders are smoothed peak (blue), hemispheric (red), and pattern match (green).

for the guinea pig models, it substantially outperforms the hemispheric decoder at low SNR, whereas it showed similar performance in quiet. Interestingly, the smoothed peak decoder also outperforms the hemispheric decoder at SNR below 10 dB, both for the guinea pig and the cat models. Indeed, although this decoder performs worst in quiet, it proves more robust than the hemispheric decoder. The poor performance of the hemispheric decoder can again be accounted for by a bias problem. At high SNR, the hemispheric difference curves become shallower, which implies that ITD estimation is biased toward the center. The problem is less present for the pattern match decoder. Note that the smoothed peak decoder tends to be biased away from the center. This is simply because BDs are more represented away from the center.

Previously, we considered a simplistic model of sound propagation, in which sounds are simply delayed. In reality, sounds are diffracted by the head. A better description of this process is that sounds arriving at the two ears are two filtered versions of the original sound, with filters depending on source direction. These are called HRTFs and can be measured in anechoic chambers. We measured high-resolution HRTFs of a stuffed guinea pig in a natural posture (*Figure 6A*). It is known that diffraction produces ITDs that depend on frequency for the same source direction, with larger ITDs in low frequency (*Kuhn, 1977*). We find the same pattern in our measurements (*Figure 6B*), and the range of ITDs is similar to previously reported measurements in live guinea pigs (*Sterbing et al., 2003*). For the cat model, we used HRTFs measured in an anesthetized cat (*Tollin and Koka, 2009a*).

*Figure 6C* displays the cell responses for sounds presented at 90° azimuth, where we used the HRTFs to filter the sound in an acoustically realistic way. We then test the estimation error in azimuth, rather than in ITD, for white noise presented in quiet (*Figure 6D*). For both animals, the pattern match decoder is substantially better than the hemispheric decoder. Indeed, since the hemispheric decoder discards all information about BF and BD, it cannot take advantage of the frequency variation of ITDs, whereas the pattern match decoder does. The difference is particularly striking for the cat due to its larger head size.

## Tuning heterogeneity as information

We have argued that the hemispheric decoder performs poorly because it discards the information present in the heterogeneity of ITD tunings of the cells. We demonstrate this point in *Figure 7A* by

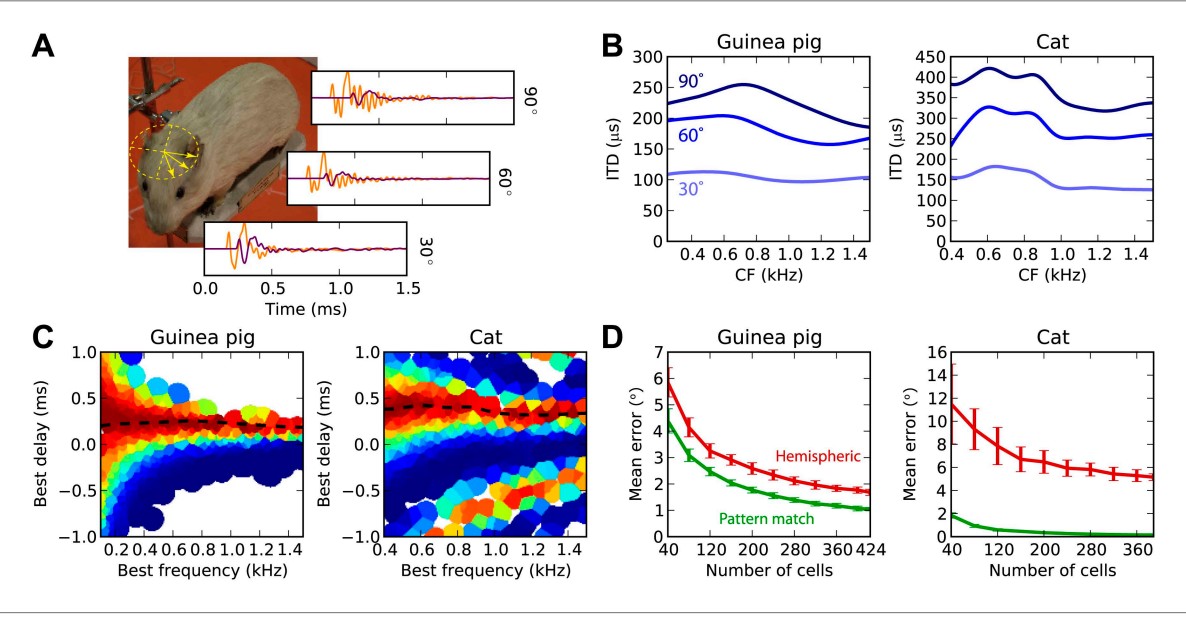

**Figure 6**. Realistic head-related transfer functions. (**A**) Photograph of stuffed guinea pig used for HRTF recordings, and three pairs of left/right ear impulse responses corresponding to the directions marked on the photograph. (**B**) Frequency dependence of ITD for the three azimuths shown in panel (**A**), in guinea pig and cat HRTFs. (**C**) Mean response of the model to white noise stimuli at the same azimuth (90°) for both animals, the frequency-dependent ITD curve is shown for this azimuth (dashed). (**D**) Performance of the model as a function of the number of cells for hemispheric (red) and pattern match (green) decoders.

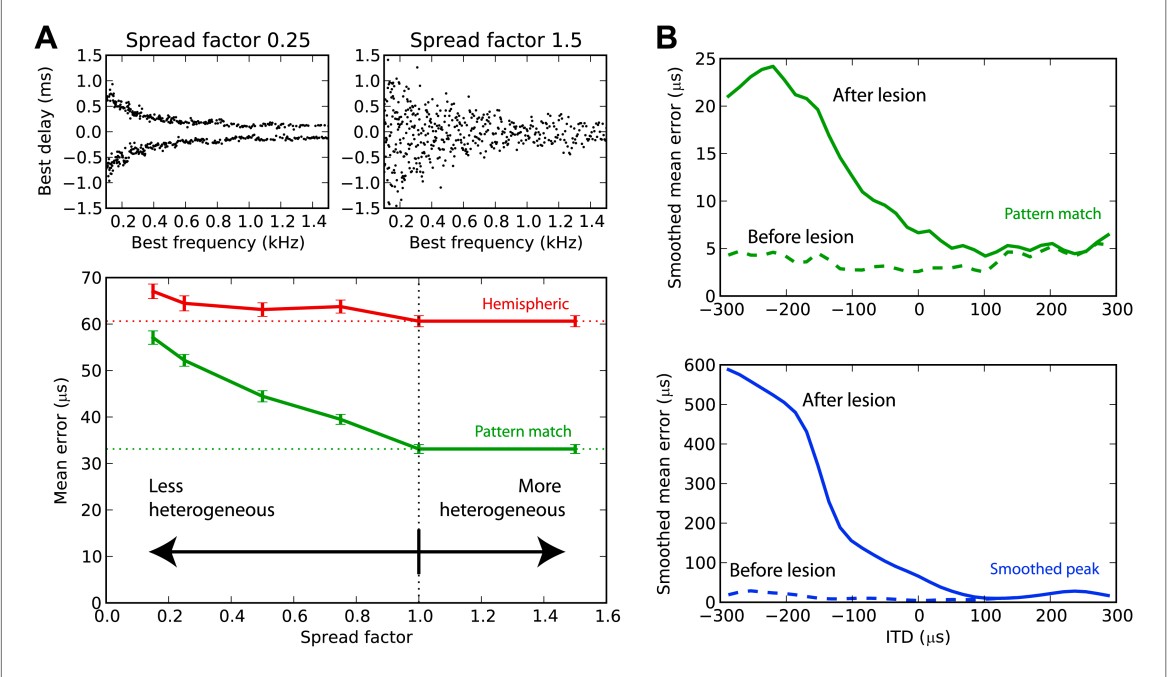

**Figure 7**. Effect of heterogeneity and lesions. (**A**) Mean error for the hemispheric (red) and pattern match (green) decoders in the guinea pig model, depending on the spread of the best delays, for white noise presented with acoustic noise (SNR between −5 and 5 dB, no HRTF filtering). For every frequency, the standard deviation of BDs is multiplied by the spread factor: lower than 1 denotes less heterogeneous than the original distribution (top left), greater than 1 denotes more heterogeneous (top right). Dashed lines represent the estimation error for the original distribution. (**B**) Mean error for the pattern match (green) and smooth peak (blue) decoders before (dashed) and after (solid) lesioning one hemisphere in the guinea pig, as a function of presented ITD. The model is retrained after lesioning. The error curves are Gaussian smoothed to reduce noise and improve readability.

varying the amount of heterogeneity in the BDs of the guinea pig model. The standard deviation of the BD is multiplied by a 'spread factor': below 1, the BD distribution is less heterogeneous than in the original distribution; above 1, it is more heterogeneous. We then test the estimation error for white noise as a function of spread factor. When the BDs are less heterogeneous, there is little difference between the performance of the hemispheric and pattern match decoder. But as heterogeneity increases, the pattern match decoder performs better, whereas the hemispheric decoder shows little change in performance. Therefore, heterogeneity of tunings is indeed useful to estimate the ITD, and pooling the responses discards this information. Performance remains stable when the distribution is made more heterogeneous than the actual distribution (spread factor >1).

## Effect of lesions

Lesion studies in cats (*Jenkins and Masterton, 1982*) and in humans (*Litovsky et al., 2002*) show that when one inferior colliculus is removed, the sound localization performance in the contralateral field drops but remains almost intact in the ipsilateral field. This is not compatible with an ITD estimation based on the comparison between the activity of the two hemispheres. We simulated a hemispheric lesion in the pattern match and smoothed peak decoders (*Figure 7B*), by removing all cells with negative BDs. The performance for positive ITDs is essentially unchanged, whereas it is highly degraded for negative ITDs, especially for the smoothed peak decoder. Lesion data indicate that sound localization performance is greatly degraded in the contralateral hemifield, but not completely abolished, which would discard the smoothed peak decoder—although lesions might not have been complete, and those were free-field experiments involving other cues than ITD.

## Owls and humans

Finally, we test the estimation performance in barn owls and humans, for white noise filtered through measured HRTFs ('Materials and methods') (*Figure 8*). For barn owls, we used a previously measured

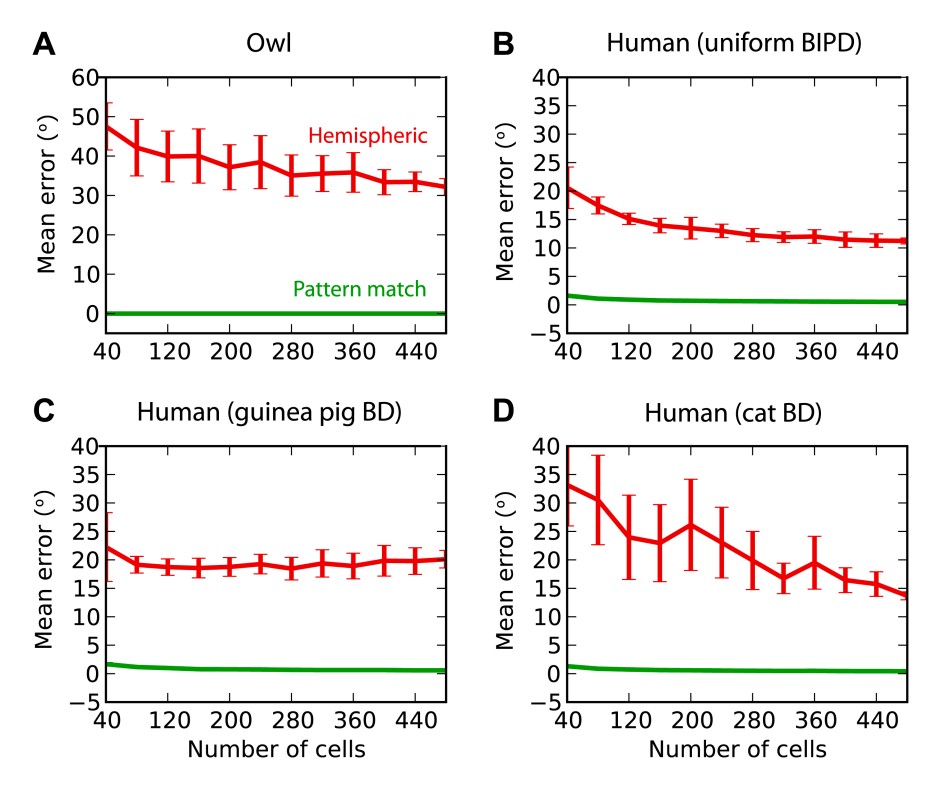

**Figure 8**. Humans and owls. (**A**) Mean error for the pattern match (green) and hemispheric (red) decoders for the barn owl model, with sounds presented through measured HRTFs. (**B**) Performance in the human model with uniformly distributed best interaural phase differences. (**C**) Performance in the human model with best delays distributed as in the guinea pig model. (**D**) Performance in the human model with best delays distributed as in the cat model.

distribution of BD vs BF (***Wagner et al., 2007***). Barn owls are sensitive to ITDs in very high frequency (about 2–8 kHz), and therefore, as expected, the hemispheric decoder performs very badly compared to the pattern match decoder (***Figure 8A***). For humans, the distribution of BD vs BF is unknown. Some indirect evidence from MRI and EEG studies suggests that BD is not uniformly distributed (***Thompson et al., 2006***; ***Briley et al., 2013***). We tested three possibilities: uniformly distributed BD within the pi-limit, similar to what is observed in birds (***Figure 8B***), BD distribution of the guinea pig model (***Figure 8C***), and BD distribution of the cat model (***Figure 8D***). In all cases, the estimation error of the hemispheric decoder is very high, an order of magnitude larger than human sound localization acuity (on the order of 3°) (***Carlile et al., 1997***).

## Comparison with behavioral performance

Our results can be compared with behavioral performance measured in psychophysical experiments. One may immediately object that the pattern decoder is in fact too accurate compared to natural performance, in particular for humans (***Figure 8***). Therefore, we must stress again that the performance obtained by a decoder in a given context is always overestimated, compared to the same decoder adapted to a more general context, 'even when tested with the same sounds'. To give an example, all decoders are much more accurate when trained and tested on a single narrow frequency band (***Figure 2B***) than when trained with broadband sounds and tested with exactly the same narrowband sounds (***Figure 3B***). Additional imprecision is introduced by other uncontrolled sources of variability in the sounds, many of which we have not considered:

1. Sound locations differ not only by azimuth but also elevation and distance, both of which impact binaural cues
2. Reflections on the ground and objects impact ITDs (***Gourévitch and Brette, 2012***)

3. There are often multiple sound sources
4. Sound sources are generally not point sources and are directional
5. Natural sounds have widely diverse spectrotemporal properties

A sound localization system adapted for the full range of ecological variability necessarily performs worse in a narrower range of conditions than a system specifically optimized for that narrow range. In addition, acoustical cues must be associated with sound location through some feedback mechanism, and therefore sound localization acuity is constrained by the precision of this feedback. Indeed a comparative study across 23 mammalian species shows that sound localization acuity is best explained by the width of the field of best vision (*Heffner and Heffner, 1992*). Therefore, our model results should be understood as a lower bound for the accuracy of these decoders. In particular, the poor performance of the hemispheric decoder is actually optimistic, all the more so as we made specific efforts to enhance it by taking into account nonlinearities (*Figure 2A*) and by applying frequency-dependent corrections (*Figure 4*).

Most psychophysical studies in animals have focused on the measurement of the minimum audible angle (MAA), which is a discrimination threshold for sources near the midline. In cats, the MAA is about 5° for broadband noises longer than 40 ms (defined as the speaker separation given 75% correct responses) (*Casseday and Neff, 1973*; *Heffner and Heffner, 1988a*). *Tollin et al. (2005)* measured accuracy in an absolute localization study in the −25° to 25° range. When the cat's head is unrestrained, direction estimates show little bias and the standard deviation is 3–4°, which corresponds to a mean unsigned error (the measure used in this article) of 2.4–3.2° (assuming normally distributed responses). In *Moore et al. (2008)*, the mean unsigned error was directly reported (although only for directions near the midline) and was in the 2–4° range. In a behavioral study in which cats were trained to walk to a target speaker in the −90° to 90° range, the animals could do the task with nearly 100% accuracy, with no apparent dependence on speaker azimuth (*Malhotra et al., 2004*)—but speakers were spaced by 15°. In *Figure 6D*, we report a mean unsigned error of 5° for the optimized hemispheric model (including frequency-dependent and nonlinear corrections; error for the pattern decoder was nearly 0°). The model was trained and tested in quiet with broadband noises, with sources constrained to the horizontal plane. Therefore, it is a very optimistic estimate, especially given that the sound localization tasks mentioned above were two-dimensional (i.e., the elevation had to be estimated as well). It could be argued that the behavioral task included additional cues, in particular interaural intensity differences, because the sounds were broadband. However, *Moore et al. (2008)* showed for sources near the midline that sound localization accuracy in the horizontal plane is very similar for broadband and low-pass filtered noises (<5 kHz), which do not include these cues.

In cats, the just noticeable difference in ITD is similar for tones of 500 Hz and 1 kHz, about 25 μs (*Wakeford and Robinson, 1974*). The performance of the pattern decoder is generally not strongly dependent on frequency in the 500–1200 Hz range (*Figures 2–4*), whereas the performance of the hemispheric decoder consistently increases with frequency (between about 500 and 1 kHz in *Figure 2*).

Unfortunately, there are no behavioral studies in guinea pigs. In the gerbil, another small mammal with low-frequency hearing, the MAA is 27° (*Heffner and Heffner, 1988b*). This makes sound localization acuity in gerbils one of the worst of all mammalian species in which it has been measured (*Heffner and Heffner, 1992*). Given that the maximum ITD is about 120 μs (*Maki and Furukawa, 2005*), the threshold ITD should be about 54 μs (using Kuhn's formula; *Kuhn, 1977*). Given that this threshold is so high, and in the absence of absolute localization studies in these two species, it is difficult to discard any model on the basis of the existing behavioral data alone. We note however that, for a given accuracy, the hemispheric decoder requires many more neurons than the pattern match decoder (*Figure 3A*).

In owls, ITD is a cue to azimuth, whereas interaural level difference is a cue to elevation (*Takahashi et al., 1984*; *Moiseff, 1989*). We found that the mean unsigned error with the hemispheric decoder was greater than 30°, when trained and tested in quiet with HRTFs (*Figure 8A*). Behaviorally, barn owls localize broadband sounds with azimuthal error smaller than about 10° at all azimuths (*Knudsen et al., 1979*), and a large part of this error is due to an underestimate of eccentric azimuths that can be accounted for by a prior for frontal directions (*Fischer and Peña, 2011*). In humans, behavioral estimates of azimuth are largely dominated by low-frequency ITDs (*Wightman and Kistler, 1992*), and the mean unsigned error with broadband noise bursts (open-loop condition) is about 5° in the frontal hemifield (2–10° depending on azimuth) in a two-dimensional absolute localization task (*Makous and Middlebrooks,*

*1990*). In contrast, the hemispheric decoder has an average error of about 10° in the most favorable scenario (*Figure 8B*), although sources are constrained to the horizontal plane.

In summary, our results with the hemispheric decoder appear inconsistent with behavioral data for cats, humans, and owls. In contrast, the results obtained by the pattern match decoder imply that there is enough information in the activity of binaural neurons to account for the sound localization accuracy of these species. There is insufficient behavioral data in the guinea pig model to distinguish between different decoders.

## Discussion

There are two major theories of ITD processing in mammals. One theory, initially proposed by *Jeffress (1948)*, asserts that ITD is represented in the activity pattern of neurons with heterogeneous tunings to ITD. A more recent theory claims that ITD is represented by the relative activity of the two MSOs, irrespective of the tunings of the cells (*Grothe et al., 2010*). We compared different ways of extracting information about sound location from the responses of a model population of binaural neurons constrained by electrophysiological data and acoustical recordings, to a large variety of sounds— which would be infeasible with single-unit recordings in animals and impossible in humans. Our results demonstrate that, although a labeled line code for ITD—the most literal interpretation of the Jeffress model—is too inefficient, summing the activity in each hemisphere discards too much of the information that is present in neural activity patterns. In addition, the heterogeneity of ITD tunings is important for decoding performance, rather than being meaningless variability (*Figure 7*). This loss of information is large enough that the hemispheric decoder cannot account for behavioral performance measured in cats and humans, although we improved it by taking into account nonlinearities (*Figure 2A*) and by applying frequency-dependent corrections (*Figure 4*). The critical flaw lies in the fact that an estimate of ITD based on global hemispheric activity is not robust to changes in sound properties other than ITD.

### Optimal coding of ITD

Our results appear to contradict the previous studies showing that hemispheric codes for ITD are optimal (*Harper and McAlpine, 2004*) and that response patterns do not provide more information than simply summing (*Lesica et al., 2010*; *Lüling et al., 2011*). However, these studies focused on a simple task, in which only the ITD was allowed to vary. This is an elementary task for a decoder because any variation in the pattern of responses can be attributed to a change in sound location. It is much more difficult to estimate location independently of irrelevant dimensions found in ecological situations, such as level, spectrum, and background noise. This point is related to the concept of 'overfitting' in statistical learning theory: an estimator may be very accurate when trained and tested with the same data, while in fact very poor when tested on new data. This is precisely what happens with the hemispheric decoder. When tested with the same sounds used to calibrate the decoder, its performance is indeed very good for the guinea pig model (*Figure 2B*), consistently with previous results. However, when the decoder is calibrated for broadband sounds and tested with the same sounds as before, performance degrades drastically (*Figure 3B*). Thus, our results directly demonstrate that indiscriminate pooling of the activity in each hemisphere is a poor way to decode information about sound location. *Figure 7A* also directly contradicts the claim that the optimal code for ITD consists of two populations of identically tuned neurons (*Harper and McAlpine, 2004*). On the contrary, heterogeneity of tunings is critical for robust estimation, consistently with theoretical arguments (*Brette, 2010*).

The discrepancy with previous arguments in favor of hemispheric or 'slope' coding seems to stem from a confusion between accuracy and acuity (*Heffner and Heffner, 2005*; *Tollin et al., 2005*): accuracy measures how well one can estimate the correct value (absolute localization); acuity measures how easily one can distinguish between two values (discrimination). Acuity can be directly related to ITD sensitivity of neural responses (favoring a 'slope code'), but accuracy is in fact the relevant ecological concept for the animal.

It may be objected that the focus on optimality may be irrelevant because animals only need to be accurate enough, given the ecologically relevant tasks. However, our results also imply that for a given level of accuracy, the hemispheric decoder requires many more neurons than the pattern match decoder (*Figures 3A and 6D*), and therefore it is energetically inefficient. Although there may be little evolutionary pressure for very accurate sound localization in some species, the same argument does not apply to

energy consumption in the brain (*Attwell and Laughlin, 2001*). Nevertheless, there are also physiological constraints on the way information can be extracted, which we examine in the section 'Physiological mechanisms' below.

## Correlations and coding

It is known that the structure of neural correlations can be critical to optimal decoding. There are different sources of correlations: anatomical divergence (overlap in the sets of presynaptic neurons), shared variability due to feedback or lateral connections, and stimulus-dependent variability (e.g., changes in level). In the medial superior olive (MSO), the earliest nucleus with ITD-sensitive neurons, correlations due to anatomical divergence are probably limited because frequencies are narrowly tuned in these binaural neurons and receive inputs from few monaural neurons (*Couchman et al., 2010*). The second source of correlations is also likely to be weak because there are no identified lateral connections within the MSO and little evidence of feedback connections, although GABAergic receptors have been recently characterized (*Couchman et al., 2012*). Thus, neural correlations in this early sound localization circuit are presumably mainly due to the shared acoustic stimulus.

When these correlations are neglected and neurons are assumed to fire independently, conditionally to the ITD, then the optimal code is the most sensitive one (*Harper and McAlpine, 2004*), and reliable estimates can be obtained by simply pooling the estimates obtained from individual responses. This conclusion is wrong in general when there are stimulus-dependent correlations (*Brette, 2010*). In this case, as we have shown, the structure of neural correlations contains useful information that can be exploited by simple decoders. For example, changes in various aspects of the sound induce shared variability in individual responses, which results in the same variability in pooled responses, but little variability in the relative activity of neurons (used by the pattern decoder).

Another mechanism to reduce the impact of shared stimulus-dependent variability is divisive normalization (*Carandini and Heeger, 2011*). Level normalization was in fact included in all the models we tested, so as to focus on ITD cues (rather than interaural level differences).

## Pattern decoders

Previous studies have assessed the performance of pattern decoders in estimating sound location from the responses of ensembles of cortical neurons. These decoders included artificial neural networks using spike counts or relative spike timing (*Furukawa et al., 2000*; *Stecker et al., 2005*; *Lee and Middlebrooks, 2013*) and maximum likelihood estimation (*Miller and Recanzone, 2009*), which is close to the pattern decoder used in this study. It is generally found that good performance can be achieved provided that the set of neurons is large enough. A previous study also found good performance with an opponent channel model, similar to the hemispheric model we tested in our study (*Stecker et al., 2005*). However, these studies tested the decoders on responses to a single type of sound (white noise), although with several levels, and as we have shown, substantial differences between the performances of decoding mechanisms only arise when sounds are allowed to vary in dimensions other than the dimension being estimated (e.g., spectrum).

In a recent experimental study, a maximum likelihood decoder was found to outperform a hemispheric decoder in estimating sound location from responses of neurons in the inferior colliculus (*Day and Delgutte, 2013*), which is consistent with our study. However, as in previous studies, only responses to a single type of sound were used, which implies that performance in more realistic scenarios was overestimated.

## Physiological mechanisms

The pattern decoder is essentially a perceptron (*Dayan and Abbott, 2001*): spatially tuned neurons are formed by simply pooling neural responses with different weights, and then the maximally active neuron indicates source location. These weights reflect the average activity pattern for the preferred location, and thus could be learned by standard Hebbian plasticity mechanisms. The smoothed peak decoder is essentially the same, except the weights are not learned. In the cat model, most low-frequency neurons in the central nucleus of the inferior colliculus are spatially tuned, with preferred azimuth homogeneously distributed in the contralateral hemifield (*Aitkin et al., 1985*). These neurons receive excitatory inputs from ITD-sensitive neurons in the MSO. In addition, when one inferior colliculus is removed, sound localization performance drops only in the contralateral field. Both the pattern and smooth peak decoders are in line with these findings.

The hemispheric decoder pools neural activity from each hemisphere, and then calculates the normalized difference, which are both simple operations. However, two remarks are in order. First, contrary to the other decoders, the presence of both functional hemispheres is required to estimate sound direction in either hemifield. Second, these operations produce a graded estimate of sound direction, not a spatially tuned response. Therefore, producing spatially tuned responses requires an additional step, with neurons tuned to a specific ratio of hemispheric activity. The firing rate of these neurons must then depend nonmonotonically on the activity of each side. Thus, the hemispheric decoder appears more complex, in terms of neural circuitry, than any of the other decoders. It could be argued that creating spatially tuned responses is in fact not necessary for sound localization behavior. For example, movements toward the sound source could be generated with activity in the two hemispheres controlling opposite muscles (*Hancock and Delgutte, 2004*). However, in addition to the fact that there are spatially tuned neurons in the inferior colliculus (*Aitkin et al., 1985*), this idea does not fit with what is known of the physiology of eye movements. Cats orient their gaze toward a briefly presented sound, a behavioral response that has been used to measure sound localization accuracy (*Tollin et al., 2005*). Eye movements are controlled by neurons in the superior colliculus (SC), which form a map (a 'place code'): stimulation of neurons in the SC produces saccades whose amplitude and direction depend on the site of stimulation, but not on intensity or frequency of stimulation (*Sparks and Nelson, 1987*). Some of these neurons are tuned to sound location (*Populin et al., 2004*).

The anatomy and physiology of the ITD processing pathway are very similar across mammalian species (*Grothe et al., 2010*). However, while hemispheric decoding might be consistent with behavioral data in small mammals, it is not with data in cats and humans. Therefore, if there is a common mechanism for ITD processing in mammals, it cannot be based on pooling neural activity on each hemisphere. A traditional argument in favor of the hemispheric model of ITD processing or 'slope coding' is that in small mammals, there are many binaural neurons with large BDs, which is contradictory with a labeled line code. However, it should be noted that the exact symmetrical argument applies as well: there are many binaural neurons with small BDs (within the ecological range), both in small and large mammals, which is contradictory with the slope coding hypothesis.

## Experimental predictions

Traditionally, physiological studies of ITD processing have focused on the question of sensitivity: how responses vary along the dimension to be estimated (ITD), which is typically measured by recording ITD selectivity curves. Indeed, there can be no information about ITD in responses that are insensitive to ITD. But sensitivity is only a necessary condition. To understand how ITD is extracted, one must identify those aspects of neural responses that are specific to ITD. In other words, one must analyze not only what varies with ITD but also what is invariant when properties other than ITD vary.

This point is related to the difference in behavioral studies between acuity, a measure of discriminability between stimuli, and accuracy, a more ecologically relevant measure of how well the animal can reach a target (*Heffner and Heffner, 2005*; *Tollin et al., 2005*). Indeed, the computationally challenging task for a sensory system is not to discriminate between two signals, but to extract meaningful information in face of the tremendous diversity of sensory inputs in ecological environments.

Such an analysis requires recording neural responses to a large variety of sounds. For practical reasons, we based our study on model responses. In principle, the same analysis could be done experimentally by recording the responses of a large number of cells to a broad set of sounds and levels presented at different locations. Given the large number of stimuli, such a study might require imaging or multielectrode recordings. An initial approach could be to look for invariant properties in the response of a subset of neurons to natural sounds presented at the same spatial location.

## Materials and methods

### Response model

The basic model consists of the following stages, illustrated in *Figure 1B*.

A sound S has a location θ, which could be an ITD or azimuth. Sounds are either (i) white noise, (ii) band-passed white noise, or (iii) colored noise with a $\frac{1}{f^\alpha}$ spectrum with color parameter α between 0 and 2 (0 = white noise, 1 = pink noise, 2 = brown noise).

The signal received at the two ears is this sound transformed by a pair of HRTFs for that location. We consider two HRTF models: (i) no diffraction, that is, frequency-independent ITDs, and (ii) HRTFs measured in an anechoic chamber (*Figures 6 and 8*). In addition to the target sound, each ear can receive an acoustic noise (*Figure 5*).

The signal received at each ear is then monaurally filtered by a gammatone filter bank (*Glasberg and Moore, 1990*; *Slaney, 1993*) with center frequencies and bandwidths defined by the animal model (see below). The equivalent rectangular bandwidth (ERB) Q factor of a filter is defined as $Q_{ERB}(f) = \dfrac{f}{ERB(f)}$ where ERB($f$) is the width of a rectangular filter that would pass as much power as the filter (for white noise). Following *Shera et al. (2002)*, we use the formula $Q_{ERB} = \beta\left(\dfrac{f}{kHz}\right)^{\alpha}$, where α and β are parameters specific to the animal model (see below).

Each binaural neuron receives two monaurally filtered inputs, one from each side, with an internal delay defined by the animal model. The firing rate response of the binaural neuron is given by the formula $\int(L+R)^k$, with a constant k defined by the animal model, and L(t) and R(t) are delayed and normalized versions of the gammatone filtered signals at the left and right ears at time t. The normalization factor is proportional to $\left|\int(L+R)^k\right|^{\frac{1}{k}}$, and chosen for a target maximal firing rate F of the binaural neuron. This binaural model is a generalization of two previous models that were found to produce good fits for the delay–response curves of the guinea pig (*Harper and McAlpine, 2004*) and owl models (*Fischer et al., 2008*). Here, we generalized it to include cats and humans, and checked that the delay–response curves give good fits to published data for the cat (*Joris et al., 2006*).

The result is the output response r of the binaural neuron, and the spike count is drawn from a Poisson distribution with mean r (so that r/T is the firing rate of the neuron for duration T).

All simulations were performed using the 'Brian' simulator (*Goodman and Brette, 2008*, *2009*) with the 'Brian hears' auditory periphery library (*Fontaine et al., 2011*).

## Animal models

Each binaural neuron then is specified by a BF and a BD so that the left channel is delayed by BD/2 and the right channel by −BD/2. The distribution of these parameters, as well as the bandwidths for the monaural filters, is defined separately for each animal model.

For all models, we used a range of 480 BFs ERB spaced between 100 Hz and 1.5 kHz, with the exception of the cat model with HRTFs (as the recorded HRTFs were not reliable below 400 Hz) and the owl (which uses higher frequencies for ITD processing). The firing rate of the binaural neurons was calibrated to have a peak of F = 200 Hz. As firing is Poisson, smaller or larger values would only increase or decrease neuronal noise. Parameters for all models are summarised in *Table 1*.

### Guinea pig

For the artificially induced ITD model (no diffraction), we use a maximal ITD of 300 μs (*Sterbing et al., 2003*) and a range of BDs measured from inferior colliculus of guinea pigs (*McAlpine et al., 2001*). Given a BF, we selected a BD from a normal distribution with the measured mean and variance for that BF. The bandwidth parameters α and β were as given in *Shera et al. (2002)*. The binaural power k was selected to match the curves in *Harper and McAlpine (2004)*. We measured high-resolution guinea

**Table 1.** Summary of animal models

| Name | ITD source | ITD range, μs | Best delays (BD) | Best frequencies (BF) | α | β | k |
|---|---|---|---|---|---|---|---|
| Guinea pig | Artificial | ± 300 | Measured | 100–1500 Hz | 0.35 | 4.0 | 8 |
| Guinea pig | HRTF | ± 250 | Measured | 100–1500 Hz | 0.35 | 4.0 | 8 |
| Cat | Artificial | ± 400 | Measured | 100–1500 Hz | 0.37 | 5.0 | 4 |
| Cat | HRTF | ± 450 | Measured | 400–1500 Hz | 0.37 | 5.0 | 4 |
| Human | HRTF | ± 950 | Uniform within π-limit | 100–1500 Hz | 0.37 | 5.0 | 4 |
| Human | HRTF | ± 950 | Guinea pig distribution | 100–1500 Hz | 0.37 | 5.0 | 4 |
| Human | HRTF | ± 950 | Cat distribution | 100–1500 Hz | 0.37 | 5.0 | 4 |
| Owl | HRTF | ± 260 | Measured | 2–8 kHz | 0.50 | 4.3 | 2 |

pig HRTFs from a taxidermist model from the Museum of Natural History (Paris), in an anechoic chamber covered with glass wool wedges, using the same protocol and equipment as for the LISTEN HRTF database (http://www.ircam.fr/equipes/salles/listen/). Because of the impedance mismatch between the skin and the air, acoustical properties are essentially determined by the shape, not by the material inside the body. The taxidermist model is both still and in a natural posture, which makes it very convenient to measure reliable HRTFs. ITDs were found to be frequency dependent, a maximal ITD of 250 μs, consistent with previously reported measurements in live guinea pigs (*Sterbing et al., 2003*).

### Cat

For the artificially induced ITD model, we used a maximal ITD of 400 μs (*Yin and Chan, 1990b*) and a range of BDs measured from cat IC (*Joris et al., 2006*). We generated a kernel density estimate (KDE) probability distribution of BD and BF from the measured set, and then for each BF, we chose a BD from the conditional KDE distribution of BD, given the BF. The measured data used characteristic frequency (CF) rather than BF; however, in a linear model such as the one used here, these two measures are equivalent. The bandwidth parameters α and β were as given in *Shera et al. (2002)*. The binaural power k = 4 was chosen to fit the data of *Joris et al. (2006)*, although note selecting a power of k = 2 to match the guinea pig model did not significantly alter the results. For the HRTF model, we used the HRTFs recorded by *Tollin and Koka (2009b)*, which had a maximal ITD of 450 μs. These HRTFs were unreliable below 400 Hz, and so this model was restricted to be used between 400 Hz and 1.5 kHz.

### Owl

We used HRTFs and a distribution of BDs measured from barn owl IC from *Wagner et al. (2007)*. BDs were chosen using the same procedure as in cats, with KDE estimates. The HRTFs had a maximal ITD of 260 μs. The bandwidth parameters α and β were as given in *Köppl (1997)*. The binaural power k from *Fischer et al. (2008)* was used. BFs from 2–8 kHz were used, as the owl is known to be ITD sensitive above 2 kHz (*Coles and Guppy, 1988*), and the HRTFs were only accurate up to 8 kHz.

### Human

HRTFs from the IRCAM LISTEN database (http://www.ircam.fr/equipes/salles/listen/) were used. These had a maximal ITD of approximately 950 μs. As the distribution of BDs in human is unknown, we used three hypothetical distributions: (i) a uniform distribution of BDs within the pi-limit, (ii) the distribution used in the guinea pig model, and (iii) the distribution used in the cat model. Bandwidth parameters and the binaural power k were as used in the cat. We tested other binaural powers and bandwidths (including the much sharper bandwidth estimates from *Shera et al. (2002)*), but these did not significantly alter our results.

## Decoders

The decoding problem is to compute an estimate $\hat{\theta}$ of θ, given the vector of responses **r** of the binaural neurons. We define a training set and a testing set of data. The acoustical inputs can be different between the two sets, for example, training with white noise and testing with colored noise (*Figure 3D*). The training set is used to set the parameters of the decoder, and the testing set is used to compute the errors and biases of the decoder ('Analysis'). We consider the following decoders, all of which can be straightforwardly implemented with a simple neural circuit:

### Peak decoder

The naive form of the peak decoder takes $\hat{\theta}(\mathbf{r})$ to be the BD of the maximally responsive neuron. We also define a smoothed form, in which the maximum is taken with respect to a Gaussian smoothed response of **r** defined by $\sigma(\mathbf{r})_i = \sum_j \omega_{ij} r_j \big/ \sum_j \omega_{ij}$, where $\omega_{ij} = e^{\frac{-(BD_i - BD_j)^2}{2w^2}}$ and $w$ is the smoothing window width.

### Hemispheric decoder

A normalized hemispheric difference λ is computed as the difference between the sum of the responses of neurons with positive BDs and the sum of the responses of neurons with negative BDs divided by the sum of the responses of all the neurons. Mathematically, $\lambda(\mathbf{r}) = \frac{\left(\sum_{i \in I} r_i - \sum_{i \notin I} r_i\right)}{\sum_i r_i}$, where I is the set of

neurons with positive BD. The estimation $\hat{\theta}(\mathbf{r})$ is defined by inverting the average hemispheric difference $\bar{\lambda}(\theta) = E[\lambda(\mathbf{r}) \mid \theta]$, where the expectation is taken over the training data. In practice, a polynomial p($\theta$) is fitted to the data ($\theta_i$, $\lambda_i$) where $\theta_i$ is the location of training datum i and $\lambda_i$ is the corresponding hemispheric difference, and this polynomial is inverted to give $\hat{\theta}(\mathbf{r}) = p^{-1}(\lambda(\mathbf{r}))$. The degree of the polynomial was chosen to maximize performance (lower degrees fit poorly but higher degrees overfit). We also consider an enhanced version of the hemispheric model able to integrate information across frequencies, the frequency-dependent hemispheric model, where the ratio is given by $\lambda(\mathbf{r}) = \dfrac{\left( \sum_{i \in I} \frac{r_i}{f_i} - \sum_{i \notin I} \frac{r_i}{f_i} \right)}{\sum_i r_i}$,

where $f_i$ is the BF of neuron i. Most papers studying the hemispheric difference model do not take varying levels into account and therefore use an un-normalized hemispheric difference. **Stecker et al. (2005)** use the maximum rather than the sum as a normalizing factor, but this is essentially equivalent.

## Pattern match decoder

Each training datum forms a response pattern we write as $\boldsymbol{\rho}$ to distinguish from the testing response $\mathbf{r}$. We compute a similarity index for each training datum

$$\psi_j = \left( \frac{\mathbf{r}}{|\mathbf{r}|} \right) \cdot \left( \frac{\boldsymbol{\rho}_j}{|\boldsymbol{\rho}_j|} \right) = \frac{\sum_i r_i \rho_{ji}}{\sqrt{\sum_i r_i^2} \cdot \sqrt{\sum_i \rho_{ji}^2}},$$

which varies between 0 (totally dissimilar) and 1 (totally similar). This is the standard cosine similarity measure from machine learning theory (the value of $\psi_j$ is the cosine of the angle between the two vectors). The estimate $\hat{\theta}(\mathbf{r})$ is the location $\theta_j$ for the index j that maximizes $\psi_j$. We also consider a frequency-dependent pattern matching decoder in which the pattern $\boldsymbol{\rho}_j$ is broken into subvectors corresponding to frequency bands, and each subvector is normalized separately. That is, each neural response is divided by the norm of all the neural responses in the same frequency band. More precisely, assuming that the neuron indices are sorted by increasing BF, and the bands are of equal size consisting of B neurons each (i.e., the first band is neurons 0 to B − 1, the second from B to 2B − 1, etc.; we used B = 40), we compute the dot product

$$\psi_j = \left( \frac{\mathbf{r}}{|\mathbf{r}|} \right) \cdot \mathrm{bandnorm}(\boldsymbol{\rho}_j) = \sum_i \frac{r_i}{\sqrt{\sum_k r_k^2}} \cdot \frac{\rho_{ji}}{\sqrt{\sum_{k = \left( \lfloor \frac{i}{B} \rfloor \cdot B \right)}^{\left( \lfloor \frac{i}{B} \rfloor + 1 \right) \cdot B - 1} \rho_{jk}^2}},$$

where for a vector $\mathbf{x}$, $\mathrm{bandnorm}(\mathbf{x})_i = \dfrac{x_i}{\sqrt{\sum_{j = \left( \lfloor \frac{i}{B} \rfloor \cdot B \right)}^{\left( \lfloor \frac{i}{B} \rfloor + 1 \right) \cdot B - 1} x_j^2}}$, where $\lfloor z \rfloor$ is the floor function, the greatest integer

less than or equal to z. Note that this banded $\psi_j$ does not vary between 0 and 1, but the value of j that maximizes it is still used to find the best estimate of the location.

In addition to these three decoders, we tested several standard decoders from machine learning and theoretical neuroscience including linear/ridge regression, nearest neighbor regression, maximum likelihood estimators, and support vector classifiers. Data for some of these are shown in **Figure 3— figure supplement 1**. Detailed results and analysis of these decoders are not presented here, as in all cases they were outperformed by the pattern match decoder. The best of these decoders was nearest neighbor regression, which performed almost as well as the pattern match decoder. The machine learning algorithms were implemented using the scikits-learn package (**Pedregosa et al., 2011**).

## Analysis

We analyze the decoders based on their errors and biases. The error is computed as $E[|\hat{\theta} - \theta|]$, where the expectation is taken over the testing data. The bias is computed by taking a linear regression through the points ($\theta_i$, $\hat{\theta}_i$) with the restriction that the line must pass through (0, 0). The bias $b$ is given as a percentage bias toward the center from the slope $g$ of the best fit line via $b = 100\,(1 - g)$. To get a

better estimate, we compute multiple values of the error and bias over 25 different shuffles of the data, and compute the mean and standard deviation of these values over the multiple shuffles. We generate 6400 total data, and to form each shuffled set of data, we take the following steps: (i) choose a subset of the full set of cells to consider (in those analyses where the number of cells was varied), (ii) choose a random subset of the data as training data, usually 400 data, (iii) choose a nonoverlapping random subset of the data as testing data, usually 800 data. This procedure was chosen to minimize biases introduced by the random sampling while keeping total computation times to a reasonable level (total computation time on an 8-core Intel i7 desktop was approximately 1 week).

## Acknowledgements

We thank the Museum of Natural History for providing stuffed animals, Hermann Wagner for sharing measured HRTFs and electrophysiological measurements of BD and BF in barn owls, Daniel Tollin for sharing measured HRTFs of a cat, and Philip Joris for sharing electrophysiological measurements of BD and BF in cat's IC. We also thank Mitchell Day, Marcel Stimberg, Agnès Léger, and Christian Lorenzi for additional comments.

## Additional information

### Funding

| Funder | Grant reference number | Author |
| --- | --- | --- |
| European Research Council | ERC StG 240132 | Dan FM Goodman, Victor Benichoux, Romain Brette |
| Agence Nationale de la Recherche | ANR-11-BSH2-0004, ANR-11-0001-02 PSL* and ANR-10-LABX-0087 | Dan FM Goodman, Victor Benichoux, Romain Brette |

The funders had no role in study design, data collection and interpretation, or the decision to submit the work for publication.

### Author contributions

DFMG, Wrote and carried out simulations, Conception and design, Analysis and interpretation of data, Drafting or revising the article; VB, Recorded HRTFs, Acquisition of data; RB, Conception and design, Analysis and interpretation of data, Drafting or revising the article

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
