## [Decision Letter]

Thank you for sending your work entitled “Decoding neural responses to sound location” for consideration at *eLife*. Your article has been favorably evaluated by a Senior editor and 3 reviewers, one of whom is a member of our Board of Reviewing Editors.

The Reviewing editor and the other reviewers discussed their comments before we reached this decision, and the Reviewing editor has assembled the following comments to help you prepare a revised submission.

This is a modeling study comparing different ways of extracting information about sound location from the responses of a model population of binaural neurons. The conclusion of the manuscript is that, while a labeled line code is too inefficient, summing the activity in each hemisphere discards too much of the information that is present in neural activity patterns.

This is an interesting and important topic that could be of general interest. However, one of the reviewers felt that it would be nice to expand on the population decoding, and discuss the fact that they have not considered population models that do not simply sum activities (e.g.,, those with optimal weighting of the contribution of each cell) and network nonlinearities useful for marginalization of task-irrelevant information like divisive normalization. Other properties known to be critical for information loss and decoding optimality, like shared variability and interneuronal correlations, have also not been considered. The reviewers would like to see a broader coverage/discussion, such that this work can appeal to the broader neuroscience community.

Specific comments:

One of the reviewers particularly liked that the authors model the effects of a unilateral lesion. Such a lesion of their pattern-match model predicts strictly contralesional localization deficits, which is what is seen in all the animal lesion studies. In contrast, a unilateral lesion of their hemispheric model predicts bilateral deficits, which are never seen in animal studies. The reviewers would like to see that result given a little more prominence, such as mention in the Abstract.

The title should be revised to more specifically indicate that the study examines “sound location based on interaural time differences”.

The results in Figure 7 for the simulated lesions should be compared, at least qualitatively, to results for the human listeners with lesions. The two models shown in Figure 7 have very different results, so the comparison may be interesting.

In the section on “Comparison to Behavioral Performance”, the argument is made that the hemispheric difference model predicts errors that are larger than those observed in behavioral studies. In the preamble to this section, it is stated that the pattern match model has very small errors, but the numerical errors for the pattern-match model are not provided for comparison to specific species and cases, as they are for the hemispheric difference model. Despite the fact that the pattern-match model's errors will appear to be quite small, they are useful and should be provided for the reader. Although the absolute errors will be smaller than behavioral thresholds, the trends in the errors across conditions are interesting to present. One can argue that the actual system will have errors larger than this very detailed model, but the (fractional) difference between model and actual performance should be comparable across conditions. (That is, it would be bothersome if the model were 1 order of magnitude too accurate in some cases, and 3 orders of magnitude too accurate in others, such that the degradation in performance required to match behavior had to change dramatically across conditions.)

The model results presented here illustrate an interesting difference in the trend of the errors between the hemispheric model and both the smoothed peak and pattern-match models at high frequencies (e.g., Figures 2, 3 and 4). The comparison of this prediction to behavioral results deserves inclusion in the section that compares model to behavioral performance. Again, this is a consistent trend in the results that can be compared qualitatively to trends in behavioral results.

In the Discussion, the argument is made that additional processing would be required to generate spatially tuned neurons from the hemispheric model. This is an interesting point, but it is not clear what spatially tuned neurons the authors are thinking about. It would help to be more specific as to what neurons (presumably at a level higher than the IC) are being considered here (indeed, there are precious few spatially tuned neurons at higher levels, and the hemispheric model doesn't require such neurons, does it?). For example, it might be helpful to consider decoding schemes proposed for the cortex. [56] (cited) tested a hemispheric model as a sort of worst-case scenario, although several authors have seized on the hemispheric model as reality. Others have looked at decoding based on spatial tuning of individual neurons, compiled as ensembles (anesthetized cat: [19]; behaving cat: [40]; awake monkey: [48]). These experimental studies should be discussed because they do empirically what the authors are doing with their simulations.

It would also be useful for the authors to include comment on the recent paper by Briley et al., JARO 2013, Vol 14: 83-101, which supports the hemispheric model, based on EEG recordings in human.

---

## [Author Response]

*This is an interesting and important topic that could be of general interest. However, one of the reviewers felt that it would be nice to expand on the population decoding, and discuss the fact that they have not considered population models that do not simply sum activities (e.g., those with optimal weighting of the contribution of each cell) and network nonlinearities useful for marginalization of task-irrelevant information like divisive normalization. Other properties known to be critical for information loss and decoding optimality, like shared variability and interneuronal correlations, have also not been considered. The reviewers would like to see a broader coverage/discussion, such that this work can appeal to the broader neuroscience community*.

The target of this research was the ongoing debate between two major competing models of sound localization, and so in the paper we focused on these models. However, we also tested a large number of other decoders, including a large number of standard decoders from machine learning. We added a figure supplement to Figure 3 showing the results of two of these decoders, ridge regression and nearest neighbor regression. Ridge regression is similar to linear regression but has an additional penalty to large coefficients. Because of the signal correlations between cells with overlapping frequency bands, the linear regression decoder was highly subject to noise, and ridge regression resolves this issue and finds the optimal weights for each cell. As can be seen in that figure, the linear/ridge regression performed similarly to the hemispheric decoder: performing slightly better in some aspects and worse in others. We also show the nearest neighbor regression decoder performs quite similarly to the pattern match decoder, which is not surprising as it is based on a similar idea (comparison of test result to stored patterns). To keep the paper to a manageable length, we only presented results of the three decoders, which we feel makes sense as two are the major current models of ITD decoding, and the third (pattern match) performed the best of all the decoders we tried. The pattern match estimator also has the virtue that it is straightforward to implement in a neural circuit, unlike many of the methods from machine learning. We also added new paragraphs to the Results and Methods sections mentioning that we tried other decoders.

Some normalization is already present in the responses of binaural neurons (see Methods, Response Model), where level is normalized, and therefore is included in all decoders. It indeed removes some task-independent sources of variability, namely overall level and ILDs. However, the differences between decoders are not due to this normalization, since it is included in all of them. The pattern match decoder also includes divisive normalization in the calculation of similarity between responses and templates, however it has no influence on the decoder’s output because it uses a winner-take-all operation and the normalization factor is identical for all candidate templates. The hemispheric decoder also uses a normalized difference, so it is not the decisive factor in the performance differences.

We added a section about shared variability and correlations in the Discussion. The bottom line is that, because the MSO (first binaural nucleus) is an essentially anatomically feedforward and tonotopic circuit, the main source of correlations should be stimulus variability. The topic of correlations is indeed very relevant to this study, because the previous conclusion that the hemispheric code (or “slope coding”) is supposedly optimal relied on the implicit assumption that neural responses are independent, conditionally to the ITD (26). But the variability of the stimulus, which is shared, implies that a better decoder must use the structure of correlations. For example, in a Jeffress-like model, changes in the sound (e.g., level, background noise) results in changes in the activity of all neurons, but the identity of the most active neuron does not change (a theoretical argument made in [4]).

*One of the reviewers particularly liked that the authors model the effects of a unilateral lesion. Such a lesion of their pattern-match model predicts strictly contralesional localization deficits, which is what is seen in all the animal lesion studies. In contrast, a unilateral lesion of their hemispheric model predicts bilateral deficits, which are never seen in animal studies. The reviewers would like to see that result given a little more prominence, such as mention in the Abstract*.

We added this to the abstract.

*The title should be revised to more specifically indicate that the study examines “sound location based on interaural time differences”*.

We used the reviewers’ suggestion of “Decoding neural responses to temporal cues for sound localization”.

*The results in*
Figure 7
*for the simulated lesions should be compared, at least qualitatively, to results for the human listeners with lesions. The two models shown in*
Figure 7
*have very different results, so the comparison may be interesting*.

We added the following sentence: “Lesion data indicate that sound localization performance is greatly degraded in the contralateral hemifield, but not completely abolished, which would discard the smoothed peak decoder – although lesions might not have been complete, and those were free-field experiments involving other cues than ITD.”

*In the section on “Comparison to Behavioral Performance”, the argument is made that the hemispheric difference model predicts errors that are larger than those observed in behavioral studies. In the preamble to this section, it is stated that the pattern match model has very small errors, but the numerical errors for the pattern-match model are not provided for comparison to specific species and cases, as they are for the hemispheric difference model. Despite the fact that the pattern-match model's errors will appear to be quite small, they are useful and should be provided for the reader. Although the absolute errors will be smaller than behavioral thresholds, the trends in the errors across conditions are interesting to present. One can argue that the actual system will have errors larger than this very detailed model, but the (fractional) difference between model and actual performance should be comparable across conditions. (That is, it would be bothersome if the model were 1 order of magnitude too accurate in some cases, and 3 orders of magnitude too accurate in others, such that the degradation in performance required to match behavior had to change dramatically across conditions.*)

*The model results presented here illustrate an interesting difference in the trend of the errors between the hemispheric model and both the smoothed peak and pattern-match models at high frequencies (e.g.,*
Figures 2, 3 and 4*). The comparison of this prediction to behavioral results deserves inclusion in the section that compares model to behavioral performance. Again, this is a consistent trend in the results that can be compared qualitatively to trends in behavioral results*.

We understand the reasoning and we had considered including a discussion of these trends. However, we had decided not to include them for two reasons, even though they are generally in line with our conclusions.

First, it relies on an assumption that the degradation of error between the “optimal” model and behavior is a frequency-independent factor. But this assumption may not be true. In that section, we argued that there are additional sources of errors, in particular the fact that the estimation of sound location relies not only on accurate acoustical cues but also on accurate feedback, such as visual feedback. If the limiting factor is the accuracy of this feedback (which is supported by the Heffner & Heffner study), then it should not be frequency-dependent, even if the coding accuracy of acoustical cues is frequency-dependent. Therefore, we think the extrapolation is a bit speculative. What is more informative, however, is when behavior is more accurate than a model (such as the hemispheric model), which does not have this additional source of error. In this case it is fair to conclude to the model is insufficiently accurate.

The second reason is that many studies use free-field experiments, and acoustical cues depend on frequency, in amount and in reliability (both ITDs and ILDs). It would bring a potentially confounding factor to the comparisons.

However, we added a paragraph in which we compared our results with a behavioral experiment using controlled binaural tones presented through earphones (no ILD), but we feel that this aspect should not be over-emphasized in this comparison. We also added a mention of the error of the pattern decoder in the first paragraph.

*In the Discussion, the argument is made that additional processing would be required to generate spatially tuned neurons from the hemispheric model. This is an interesting point, but it is not clear what spatially tuned neurons the authors are thinking about. It would help to be more specific as to what neurons (presumably at a level higher than the IC) are being considered here (indeed, there are precious few spatially tuned neurons at higher levels, and the hemispheric model doesn't require such neurons, does it?). For example, it might be helpful to consider decoding schemes proposed for the cortex.*
[56]
*(cited) tested a hemispheric model as a sort of worst-case scenario, although several authors have seized on the hemispheric model as reality. Others have looked at decoding based on spatial tuning of individual neurons, compiled as ensembles (anesthetized cat:*
[19]*; behaving cat:*
[40]*; awake monkey:*
[48]*). These experimental studies should be discussed because they do empirically what the authors are doing with their simulations*.

We wrote in the previous paragraph: “In the cat, most low frequency neurons in the central nucleus of the inferior colliculus are spatially tuned, with preferred azimuth homogeneously distributed in the contralateral hemifield (1).” Perhaps more importantly, sound localization behavior does require that the graded code assumed in the hemispheric model be converted to spatially tuned responses, at least for eye movements, which have been used to measure sound localization accuracy in cats. We now discuss this point in the Discussion (Physiological mechanisms). In the superior colliculus (to which IC projects), there are spatially tuned auditory neurons, whose stimulation produces movements of fixed amplitude and direction, independently of stimulation strength, and they are arranged topographically.

We did not address the question of how cortical responses might be decoded, because the controversy we address arose from measurements of responses in MSO and IC, and because these are earlier in the auditory pathway. However, we agree that there are a number of papers about estimating sound location from cortical responses, which would be relevant to discuss. We added a paragraph about them in the Discussion (Pattern decoders). Some of the decoders used in these studies are, indeed, similar to the pattern decoder we used (the closest one being the maximum likelihood decoder in [48]), however we do not agree that these studies do empirically what we are doing with simulations. Our key point is that decoders should be tested in a situation when the stimulus is allowed to be variable (different spectrum, etc.); otherwise a large part of the problem is neglected (stimulus-dependent variability). The difference between hemispheric decoder and pattern decoder only appears in this more general situation. The studies cited above generally find that decoders based on patterns (either spike timing or spike counts) perform well, provided there are enough neurons, but because the stimulus is fixed (except for changes in level), they do not show that such decoders are robust (i.e., would work with other sounds). Conversely, the opponent channel model shows good results in one cortical study by Stecker et al., but as in other previous studies in subcortical areas, it may not be robust to stimulus-dependent variability other than level. Our analysis strongly suggests that it is not, because the hemispheric difference is likely also sensitive to sound spectrum and other features that cortical neurons are tuned to, and because adding more neurons in the decoder does not remove stimulus-dependent variability.

*It would also be useful for the authors to include comment on the recent paper by Briley et al., JARO 2013, Vol 14: 83-101, which supports the hemispheric model, based on EEG recordings in human*.

As far as we understand it, this paper is about cortical responses to different sound locations, not about how these responses might be “decoded” into an estimate of sound location. It provides some indirect evidence that the distribution of BDs in humans is similar to what was found with single-unit electrophysiology in small mammals, i.e., more neurons with large BDs than expected from a uniform distribution. This observation does not by itself imply that sound location is estimated from the average response (note that the behavioral part in that study is in fact an analysis of the sensitivity of the hemispheric decoder, not of its accuracy as we do in our study). In fact, in our study, we start precisely from such a distribution of BD and show that the hemispheric is suboptimal. Therefore, that paper cannot be taken as evidence in favor of the hemispheric model. The approach is also indirect, as individual neural responses are not measured, and therefore a number of factors other than BD could contribute to EEG responses (e.g., peak firing rate, tuning width, if these properties have a non-zero correlation with BD). We added a mention of that paper in the paragraph about humans.